# Fiber Bundle Image Reconstruction Using Convolutional Neural Networks and Bundle Rotation in Endomicroscopy

**DOI:** 10.3390/s23052469

**Published:** 2023-02-23

**Authors:** Matthew Eadie, Jinpeng Liao, Wael Ageeli, Ghulam Nabi, Nikola Krstajić

**Affiliations:** 1School of Science and Engineering, Centre for Medical Engineering and Technology, University of Dundee, Dundee DD1 4HN, UK; 2School of Medicine, Centre for Medical Engineering and Technology, University of Dundee, Dundee DD1 9SY, UK; 3Diagnostic Radiology Department, College of Applied Medical Sciences, Jazan University, Al Maarefah Rd, P.O. Box 114, Jazan 45142, Saudi Arabia

**Keywords:** endomicroscopy, machine learning, fiber bundle

## Abstract

Fiber-bundle endomicroscopy has several recognized drawbacks, the most prominent being the honeycomb effect. We developed a multi-frame super-resolution algorithm exploiting bundle rotation to extract features and reconstruct underlying tissue. Simulated data was used with rotated fiber-bundle masks to create multi-frame stacks to train the model. Super-resolved images are numerically analyzed, which demonstrates that the algorithm can restore images with high quality. The mean structural similarity index measurement (SSIM) improved by a factor of 1.97 compared with linear interpolation. The model was trained using images taken from a single prostate slide, 1343 images were used for training, 336 for validation, and 420 for testing. The model had no prior information about the test images, adding to the robustness of the system. Image reconstruction was completed in 0.03 s for 256 × 256 images indicating future real-time performance is within reach. The combination of fiber bundle rotation and multi-frame image enhancement through machine learning has not been utilized before in an experimental setting but could provide a much-needed improvement to image resolution in practice.

## 1. Introduction

Looking at the tissue under a brightfield microscope is one of the best ways to observe cells and subcellular features. Standard tabletop microscopes require the tissue to be extracted before it can be observed. Fiber bundle endomicroscopy enables clinicians to insert a tube into a body that allows them to see cells in vivo and in situ [1]. One of the main drawbacks of using fiber-bundle endomicroscopy, however, is the desire for the images to be reconstructed and manipulated using complex software before display. Fiber bundle endoscopes use many fibers (cores), between 5000 and 30,000, which act as sensors. The total size of the fiber bundle is usually <1.5 mm in diameter; each core is then surrounded by cladding, which prevents light from affecting the surrounding cores. Due to the spacing and the cladding around each fiber, gaps are present between the cores that need to be filled to see a full image of the tissue. These gaps are normally filled using Gaussian blurring or linear interpolation; however, machine learning (ML) provides a promising approach to go beyond classical image reconstruction and image inpainting [2].

The removal of the cladding, or the so-called “honeycomb effect”, has been an important aspect of fiber bundle endomicroscopy image processing. The cladding hides important information, causing problems in diagnosis. Many methods have been used to overcome this. For example, adaptive masks based on Nyquist-Shannon sampling were designed to improve output images [3]. A darkfield illumination configuration was also used to lower spectral reflection at the proximal end of the fiber [4]. In endocytoscopy, images were reconstructed using only Gaussian blurring to remove the honeycomb pattern [5]. Histogram equalization prior to Gaussian smoothing was also used, resulting in the removal of the honeycomb effect and improving the contrast-to-noise ratio (CNR) 3-fold [6].

A more common method for image reconstruction is linear interpolation. This is completed by measuring the intensity of each fiber and interpolating between fiber core centers to fill in the unknown values using the measured intensities. Detecting core locations is a computationally costly process, which can be sped up using a lookup table to provide information, allowing for real-time interpolation [7]. By comparing different strategies of interpolation, it was found that methods using local differential properties are only slightly better in accuracy but require more computational power, concluding that they can be a good choice when smoothness and continuity are the focus over computation time [8]. However, linear interpolation is not without its drawbacks, mainly the artefacts that come with drawing lines between the cores and generating triangular patterns [9].

Machine learning has provided impressive results in fluorescent microscopy [10,11,12]. More recently, groups have been investigating the use of ML as a new method of reconstructing fiber bundle images. Single-image super-resolution (SISR) is a method of using a single-image as input and attempting to produce a super-resolved image. This is completed by learning similarities between low resolution and ground truth to produce weights. Examples of such networks include Exemplar-based Deep Neural Networks (DNNs) that train on realistic synthetic images to produce high-resolution images [13]. It is believed to be possible, due to the lack of ground truth images, that a model can be trained using synthetic images but applied to real scenarios and still produce improved outputs [13]. A Convolutional Neural Network (CNN) combined with a custom layer using Nadaraya-Watson kernel regression has produced improved results when training on irregularly sampled sparse data [9]. A dense network, which can be enhanced with the use of skip connections to reduce the effect of overfitting when using large datasets [14]. Recent advances in endomicroscopy include a generative adversarial restoration neural network (GARNN) [15]. In this implementation, the optical enhancement is obtained from training on pairs of raw fiber bundles and ground truth images captured using a dual-path imaging system.

The less commonly used method of ML is multi-frame super-resolution (MFSR), in which multiple images are used as input to produce a single super-resolved image. The drawback is that it requires multiple images of the same scene, but this is entirely feasible using fast machine vision cameras. MFSR was used by Y. Huang et al. to improve the quality of images by 1.77-fold compared with SISR [16]. Prior work in applying multi-frame ML in endomicroscopy was performed by J. Shao et al. [17]. They start with a single frame method using a dual path system to image (a) the ground truth and (b) fiber bundle images at the same time, the same as their single-image experiment [15]. In addition, they added an XY stage onto the fiber bundle path, enabling them to issue random movements during imaging to replicate clinically observed motion artefacts. Thus, by training a neural network to align the images and another to enhance the images, one can benefit from motion artefacts. When using machine learning, multiple aspects need to be considered, such as the model itself and the number of training images, etc.

Endomicroscopy can be used to image the gastrointestinal tract [18], Barrett’s esophagus [19], colorectal cancer [20], gastric cancer [21], urinary tract [22], and lung [1], among many other areas [23]. One of the key features to determine when tissue is cancerous is nuclei size [24]. To determine the nuclei’s size, the tissue needs to be broken down into four parts: the stroma, nucleus, cytoplasm, and lumen [25], which can easily be completed using segmentation methods such as k-mean clustering [26]. The size of nuclei in the segmented region can be easily measured. It can be found that cancerous regions have larger nuclei compared with non-cancerous regions [27].

In this paper, we present a multi-frame algorithm that carries out end-to-end reconstruction and enhancement of synthetic fiber bundle images using machine learning by combining two previously used methods: machine learning and fiber bundle rotation [28]. These methods, however, have not been used in conjunction and provide great benefits to the output images. The synthetic fiber bundle images are reconstructed to a higher quality than linear interpolation, as well as our single-image ML model, revealing more information about the underlying tissue. Furthermore, the enhanced images can be used to produce a more accurate nuclei size, and this could aid in the diagnosis of certain illnesses where cell nuclei size is affected and normally would require invasive tissue extraction to see cells clearer.

## 2. Methods

Due to the nature of fiber bundle imaging in vivo, inherent motion caused by breathing creates artefacts during imaging. This, paired with a high framerate, means that images of the same area are captured with minor shifts; these small shifts have the effect of oversampling. By pairing multiple images of the same area, more information is available for training, making it perfect for multi-frame super-resolution. Tissue request approval was obtained from the Tayside tissue bank, University of Dundee, Ninewells Hospital, and Medical School, Dundee, UK, along with institutional approval (approval number IGTCAL5626). The histopathology of prostate cancer tissue following radical surgery was investigated. The H&E-stained slides of the resected tissue were prepared using standard methods. An experienced histopathologist identified the cancerous and non-cancerous areas in the slides and marked them for endomicroscopic analysis. These slides were also used in experiments using second harmonic imaging previously reported [29,30].

Our method was inspired by ref. [31], in which they combined satellite images using multi-frame super-resolution to increase the resolution and remove random artifacts such as clouds to see a clearer image of the ground behind it. This is similar to fiber bundles that certain features of tissue are covered by cladding (clouds) that can be removed with multiple images. We combine this method with the method used by ref. [28], in which they were able to improve the resolution of fiber bundle images by rotating the tip of the fiber to expose the areas hidden by cladding. This rotation method was tested on USAF targets and mouse brain samples. The images were combined using the Fourier transform method along with low-pass filters. The rotations in this method were in 45-degree increments, which is too large for in vivo operations. In our method, the rotations are reduced to 2-degree increments between −10 and +10 degrees. Rotating the fiber bundle fast and at large angles in vivo will increase the risks of damage to the fiber bundle. Therefore, a small rotation of 20 degrees was chosen. Rotation enables oversampling the cladding between the fiber cores, which is naturally small, and so small rotations offer a greater probability of sampling these areas. Furthermore, as the fiber bundle has non-uniform spatial resolution due to variation in core size—rotation is bound to improve spatial resolution uniformity. One disadvantage of circular rotation is that the central regions will be poorly sampled compared with the outer regions and thus will not be as greatly impacted by this technique. However, this method does provide the benefit of improving spatial resolution without the addition of translational mechanisms on the tip, which would constrain the size of the tip.

### 2.1. Image Acquisition System

A custom-built system (Figure 1) based on an endomicroscopy system was used along with custom software to image the prostate slide with cancerous regions. The slide was placed on an XY stage so it could be maneuvered to image multiple areas. A white light was used to illuminate the sample, which then entered an x20 objective lens. The light then travels through a tube lens (TTL200-S8) and is imaged by an sCMOS camera (Hamamatsu Orca flash 4.0, Hamamatsu, Shizuoka, Japan). The image acquisition software was programmed in C++ using the Qt framework version 5.14.2 and OpenCV version 4.5.1.

To determine the pixel resolution of the system, we use a USAF target. The target was imaged, and the lines in group 4 element 1 were measured using the ImageJ line profile tool. Each line in 4:1 has a width of 31.25 µm, and across the three lines, the average pixel count was 99 pixels, giving a pixel resolution of 0.316 microns per pixel (Figure 2).

### 2.2. Multi-Frame Auto Encoder (MFAE)

An autoencoder, CNN, was chosen as the model for this work. Our model has been designed with a minimal number of layers to reduce training time due to computer specs and provide a pathway to real-time imaging in the clinic (e.g., 10 frames per second). Deep networks often have a problem with overfitting [14]. MFSR was chosen as it complements naturally with clinical widefield endomicroscopy where tissue motion is inherent and the cameras used are fast with frame rates exceeding 200 fps over USB3 or Gigabit Ethernet (e.g., see the Grasshopper3 family from FLIR, UK) [32].

The loss function used is the sum of MAE $\left(L_{1}\right)$, MSE ($L_{2}$), and SSIM as optimisers and are defined as:(1)$$\mathrm{MAE}=\frac{1}{N}\sum_{i=1}^{N}\left|y_{i}-x_{i}\right|$$
(2)$$\mathrm{MSE}=\frac{1}{N}\sum_{i=1}^{N}{\left(y_{i}-x_{i}\right)}^{2}$$
(3)$$\mathrm{SSIM}\left(x,y\right)=\frac{\left(2\mu_{x}\mu_{y}+C_{1}\right)\left(2\sigma_{x,y}+C_{2}\right)}{\left(\mu_{x}^{2}+\mu_{y}^{2}+C_{1}\right)\left(\sigma_{x}^{2}+\sigma_{y}^{2}+C_{2}\right)}$$
(4)loss=0.001∗MAE+MSE+0.001∗SSIM
where *y* is the ground truth (a high-resolution image), and *x* is the predicted output image. In SSIM $\mu_{x}$, $\mu_{y}$ are the mean intensities of *x*, *y*; $\sigma_{x}$, $\sigma_{y}$ are the square roots of the variances of *x*, *y*; and $C_{1}$, $C_{2}$ are constants to avoid instability [33]. Comparisons of loss functions, i.e., only using MSE, MAE, and MSE + SSIM, were conducted, and results showed the combination of all three gave the best results.

The network was programmed in Python 3.9, utilizing the open-source deep-learning library TensorFlow v2.6. The model is an encoder-decoder that receives the stacked image with all its segments (C) and expands the number of channels (represented by the width of the block) to extract features, then decreases the number of channels to 1 and exports a grayscale super-resolved image (Figure 3). The model is optimized using the adaptive moment estimation (Adam) optimizer with an initial learning rate of $1\times 10^{-4}$. The loss parameters of the model were set as follows: MAE weight = 0.001, MSE weight = 1, and SSIM weight = 0.001. While these values were set after empirical evaluation, we followed these broad guidelines: (i) MSE weight is the largest, so we can reconstruct a higher resolution image from a lower quality image [34]; (ii) MSE alone can result in blurred images, so we adopt two further weights, MAE and SSIM [34]. The exact values used were kept constant throughout this study.
(1)Conv + ReLU: Convolutional layer with filter size of 3 × 3 with a ReLU activation function and “same” padding to keep spatial resolution the same. The width of each block in Figure 3 denotes the number of features.(2)Max Pooling: a pooling layer, with pool size of 2 × 2, is used to reduce the spatial size of the features to remove unnecessary information and helps to reduce overfitting.(3)Up-sampling: this layer is used to decrease the number of feature maps while also increasing the spatial size of the images back to their original size.

Initial results were tested on a lab computer with an Intel Xeon W-2104 3.2GHz CPU, 32GB RAM, and an NVIDIA Quadro P2000 GPU. These specs were enough to train a simple model with a small dataset of 210 images. Only later was the computer upgraded to allow for the 2100 images in the final dataset. Initial training was completed in 100 epochs, with each epoch taking around 13 s to complete with our hardware. Later training and testing are performed on a PC with an Intel Xeon W-2195 2.36GHz CPU, 64GB RAM, and an NVIDIA Quadro 6000 GPU, with each epoch taking around 8 s.

### 2.3. Synthetic Image Generation

First, 100 images (80 for training and validation, and 20 for testing) were acquired from a single slide using the custom-built setup (Figure 3) at a resolution of 2048 × 2048. This was completed by scanning over the slide sample using an XY stage to avoid overlapping images. To maintain the robustness of the model, no specific features were targeted during image sampling. Due to the nature of the slide sample having an uneven surface, the outer areas were out of focus and needed to be removed. These images were then sliced into 256 × 256 segments which were used as the ground truth for the fiber images, resulting in 1680 images. A binary mask created using a light field image from a Fujikura FIGH-30-650S fiber bundle (Fujikura, Tokyo, Japan) is placed over the images; this introduces the undesired cladding effect. This fiber bundle has a total of 30,000 fiber cores with an average core size of 3–4 µm. We assume contact imaging is deployed, which is common practice in fiber bundle imaging as it enables optimal focus. Contact imaging ensures fiber bundle is in direct physical contact with the underlying tissue. The lateral resolution is therefore the same as the size of the fiber cores (3–4 µm). Each core is then averaged to represent a single-pixel detector to better simulate the image as seen through a real fiber-bundle. This bundle is rotated 11 times in two-degree intervals from −10 degrees to +10 degrees from normal, making 1680 stacks of 11 images for a total of 18,480 individual images.

These stacks are then combined into an array to create a 1680 × 256 × 256 × 11, grayscale array of images. The low-resolution (LR) images can be considered in perfect alignment with the high-resolution (HR) ground truth images, which give the best circumstances for reconstruction as there are no motion artifacts.

The machine learning model uses the training dataset to learn weights between the LR and ground truth images. These weights are applied to the validation dataset to determine if the output is comparable to the ground truth using the loss function (Equation (4)). If the loss does not change in five epochs, then the model will stop training. Finally, this trained model can be applied to the testing dataset to create SR images based on LR images that it has not seen during the training stage. A comparison of different numbers of rotations was completed to determine the most optimal number of rotations to perform to maximize the SSIM of super-resolved images (Figure 4). Figure 4 shows the SSIM of a group of 21 super-resolved testing images with the standard deviation of each group. The graph shows that 11 images give the highest SSIM with the lowest standard deviation, but more interestingly, it shows that at 15 rotations, there is a lower SSIM. A lower number of rotations improves the potential for clinical translation and real-time performance. For example, the standard camera used in endomicroscopy can image at 100 fps with 11 rotations, which could potentially give an ML-resolved image framerate of 9 fps.

### 2.4. Single-Image Autoencoder (SIAE)

For a comparison of multi-frame to single-image super-resolution, the algorithm described in Section 2.2 was adjusted to receive only a single-channel input image, the steps in the architecture remained the same, and a single output image was generated. Single-image architectures are only able to obtain information from a single input image; however, they will take less time to train and may still be comparable to linear interpolation. The dataset generation for the single-image is kept the same for continuity. Each of the 11 rotated fiber bundles are input as single-channel images rather than multiple channels to give a stack of 18,480 × 256 × 256 × 1.

### 2.5. Nuclei Diameter

To determine the diameter of nuclei, an image can have a threshold value applied to it, turning all pixels below the threshold black and all pixels above white. For the images acquired, it was determined that a threshold value of 100 removed all other information other than the nuclei. The diameter of each nucleus could then be measured by counting the number of black pixels across the nuclei. ImageJ was used to measure the diameter of multiple cell nuclei for comparison to ground truth and linear interpolation.

## 3. Results

First, we evaluate our model by comparing it to Gaussian blurring, linear interpolation, and our single-image variant. We used peak signal-to-noise ratio (PSNR) and structural similarity index measurement (SSIM) to objectively assess images [33]. However, they are defined by pixel differences and thus may not reflect the true image quality [35].

PSNR is an image evaluation method that calculates the ratio between the signal and noise of an image in decibels. It relies heavily on the pixel-to-pixel comparison, meaning that slight differences in pixel values may cause the PSNR value to decrease. Therefore, it’s values may not reflect the human visual perception of the images [36]. The PSNR of the images is determined by Equation (5).
(5)$$\mathrm{PSNR}=20*\log_{10}\left(\frac{255}{\sqrt{\mathrm{MSE}}}\right)$$

MSE is determined by Equation (2). SSIM is an image perception-based method based on multiple variables, such as luminance, contrast, and structure. Therefore, it is more suitable for image interpretation and gives a better representation for image enhancement. The equation for SSIM is stated above in Equation (3).

Figure 5a presents the ground truth images as taken from the microscope setup. Figure 5b is a synthetic recreation of the same scene as imaged using a fiber bundle. Figure 5c–f shows the reconstruction of the fiber image using Gaussian blurring, Linear interpolation, single-image autoencoder, and our proposed multi-frame auto-encoder, respectively. Red arrows have been drawn on each of the images to direct attention towards a small cell nucleus that is difficult to identify in most cases except MFAE.

Analysis shows that the SSIM is increased from an average of 32% in SIAE to 83% in MFAE (Table 1), and features appear better defined, as indicated by the red arrow in Figure 5. Having multiple rotated images to perform MFSR was not only capable of removing the honeycomb effect and restoring the information hidden by the cladding, but also the mean SSIM of fiber bundle images improved by 1.97-fold compared with the linear interpolation method on a single-image. Further work could improve the quality of the images but would require a much more complex architecture. From 420 testing images, the PSNR and SSIM were computed for each image (Table 1). When compared with similar attempts in multi-frame endomicroscopy, our method is the fastest (30 ms vs. ~1 s for [15,17]) with a higher SSIM and comparable PSNR.

Note in Table 1 that the PSNR of MFAE appears lower than that of other methods; this may be due to several reasons, such as the contrast or edges being sharper than those of the original, which the evaluation method would determine to be noise.

Rotating the fiber bundle provides more information on the same tissue by covering more of the image with fibers and concatenating images together. The method of rotating the fiber bundle was suggested by ref. [28], but the images were only compared using signal-to-noise ratio and contrast-to-noise ratio (CNR); to our knowledge, no one has used this method to analyze nuclei size on fiber bundle images.

To check the resilience of our network, a follow-up experiment was carried out that used different synthetic fibers, which the model had not seen, and applied the previously trained model to the new datasets. While the output of the resolved images was not of the same quality as those made using the fiber bundle it was trained on, the SSIM was still higher than that of linear interpolation and SIAE methods. However, with more rigorous training and post-processing the SSIM value could be similar to that of the original dataset.

To further test the resolution limits of the MFAE model, a modulation transfer function (MTF) was performed using images of sinewaves. By using increasing frequencies of sinewaves until the model could no longer reconstruct the wave visibly, the results of this test allowed the model to reconstruct waves down to 9 pixels per sinewave (0.316 microns per pixel) or a frequency of 356 cycles/mm. Meanwhile, linear interpolation was only able to reconstruct waves up to 19 pixels per sinewave, or a frequency of 169 cycles/mm. This gives our model a resolved resolution of around 3 µm compared with that of linear interpolation, which has a resolution of 6 µm.

### 3.1. Nuclei Size

To obtain an improved representation of the tissue post-reconstruction, the testing images were stitched together (Figure 6). Figure 6a is the original ground truth image. Figure 6b has been reconstructed using linear interpolation. This method is not accurate as the borders of each section are missing due to the nature of linear interpolation. Lastly, Figure 6c shows the multi-frame super-resolution images. Red lines indicate the cell nuclei to be measured and compared; this helps determine the accuracy of super-resolved shapes. All measurements were taken across the same area.

To measure the nuclei more accurately, the images are first thresholded at a value of 100, which was determined to incorporate just the cell nuclei. The result of the measurements is shown in Table 2. From the results in Figure 6, the super-resolution nuclei are similar in size to the ground truth nuclei, whereas the nuclei in the interpolated images are much smaller. The sizes measured in Table 2 are an estimate made by multiplying the pixel size by the pixel resolution (stated as 0.316 µm in Section 2.1); these values do not reflect the actual resolution of the system. Fiber bundle endomicroscopy in contact imaging is not a diffraction-limited system (which would result in a resolution of 500 nm or less) because of the size of the cores. However, multi-frame sampling allows us to break the Nyquist limit imposed on single-frame analysis by taking more samples across a range of angles. The diffraction limit remains, and we have not reached it with this study. Prostate cells can have a diameter between 5 µm and 12 µm [37], and therefore our results are in line with typical prostate cell sizes.

### 3.2. Computational Performance

The average execution time of the super-resolution model was 0.03 s for images of size 256 × 256. The real-time application of the model could be possible, and if the number of input channels is reduced, a higher framerate could be achieved. The current use of 11 rotated LR frames from a single cropped sample means that more images of the same area need to be captured before the algorithm can be used, and the time to reconstruct SR images is high. The solution to this problem would be to use fewer images, this would reduce the time for an SR image to be produced; however, with fewer images, less information is available for the model to train, which would reduce output quality (Figure 4).

A more complex model along with fewer LR images may provide a more desired outcome with faster enhancement times. Training the model would take a significantly longer amount of time.


*The process of fast fiber bundle imaging makes MFSR a more viable option when imaging tissue with a desire for high-resolution images. The requirement of multiple images does, however, remove the possibility of real-time use of SR during imaging, depending on the number of frames used during the MFSR process. It does provide the ability to perform cell nuclei measurements with higher accuracy than linear interpolation.*


### 3.3. Working towards Real Fiber Bundle Imaging

The model was tested on real fiber bundle images of a USAF target and reconstructed using the synthetic training discussed in this paper. Suboptimal results obtained highlighted the limitations of the current scheme and point to future work:A deeper machine learning model is needed for more realistic scenarios. The model used in this paper is small in comparison to most models used for image enhancement. However, as a proof of concept, we demonstrate improved resolution with oversampling.Minimization of rotation axis shift and rotational angle errors is required to have a system that will work in a clinic. This can be achieved in two ways: first, by imaging quickly, and second, by having a more robust machine learning model.The robustness of the training dataset could be improved by adding more variation in rotation angles by implementing random rotations within given boundaries. Using a greater number of fiber bundle masks during training would increase robustness to core pattern differences.

## 4. Conclusions and Future Work

In this paper, we present a simple MFSR model that combines two previously tested methods of fiber bundle enhancement, machine learning and fiber bundle rotation, to reconstruct synthetic fiber bundle images into images with enhanced resolution. To the author’s knowledge, these two methods have not been used in combination before and are a potential area for further advancement. We have demonstrated that rotation of the fiber bundle is a practical technique to provide multiple images that can be used for multi-frame super-resolution. The SR images produced from our model have improved PSNR and SSIM compared with linear interpolation and single-image super-resolution. Our images can then be used to calculate nuclei size, an important metric for defining cancerous cells and cancer grades.

With further work, our technique could be developed into a powerful tool for fiber bundle endomicroscopy. We plan to exploit motion artifacts present in real-time imaging to enhance images on the fly and translate it to a clinical setting [17]. While synthetic images can be used for training [13], training on real tissues of interest will result in more realistic images [17]. Further optimization and improvement could decrease training time and improve output image quality. The H&E slides provide inherently higher contrast on cellular features than autofluorescence would do in a real clinical setting, indicating that in vivo staining techniques will be desirable [38]. Lastly, a more adaptive network is needed to deal with the dynamics of the transmission properties of the fiber bundle. Each core has an associated transmission coefficient, which varies with bending, stress, strain, and temperature. The temporal dynamics of transmission coefficients were not considered in the current study and will be part of a future effort.

## Figures and Tables

**Figure 1 sensors-23-02469-f001:**
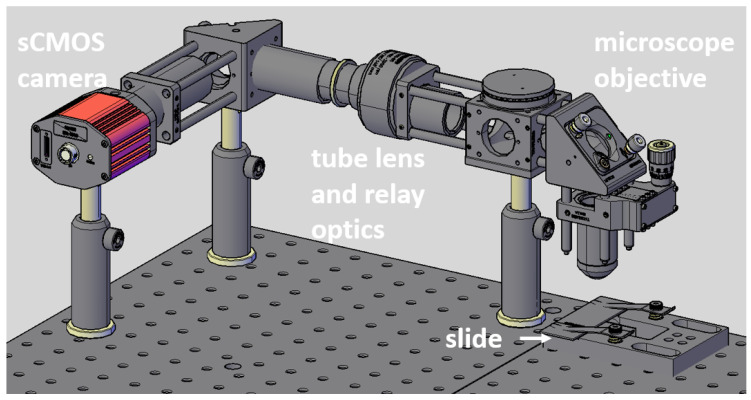
CAD model of custom setup used to image slide.

**Figure 2 sensors-23-02469-f002:**
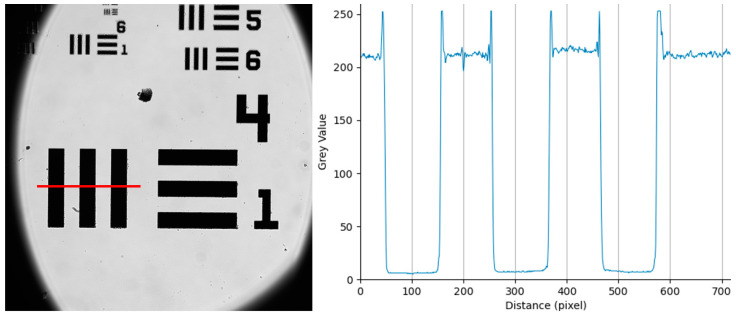
The captured image of a 1951 USAF target, measuring along group 4:1 to minimize measurement error, the average line width was 99 pixels. Group 4:1 has a line width of 31.25 µm. Red line represents line profile taken for line plot.

**Figure 3 sensors-23-02469-f003:**
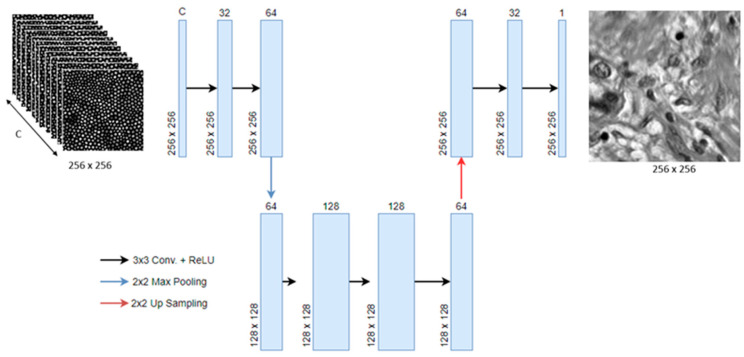
Proposed MFAE model. C represents the number of segments (channels) to be combined.

**Figure 4 sensors-23-02469-f004:**
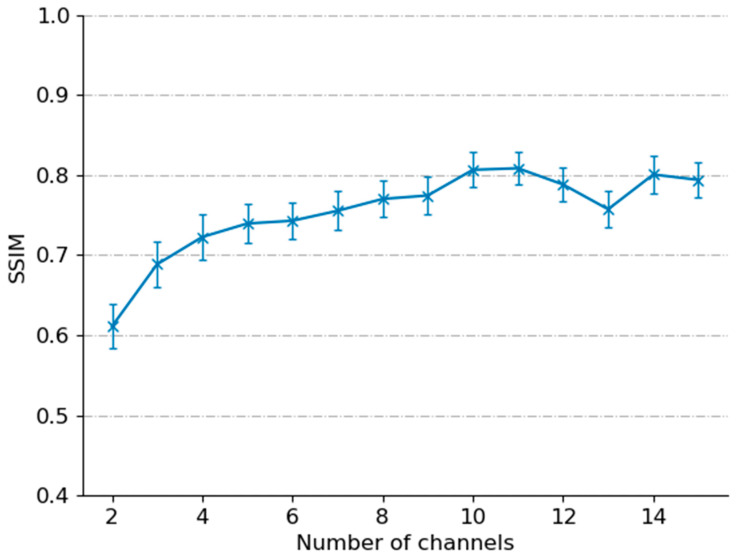
The plot of SSIM with a different number of channels. 11 channels provided the highest SSIM.

**Figure 5 sensors-23-02469-f005:**
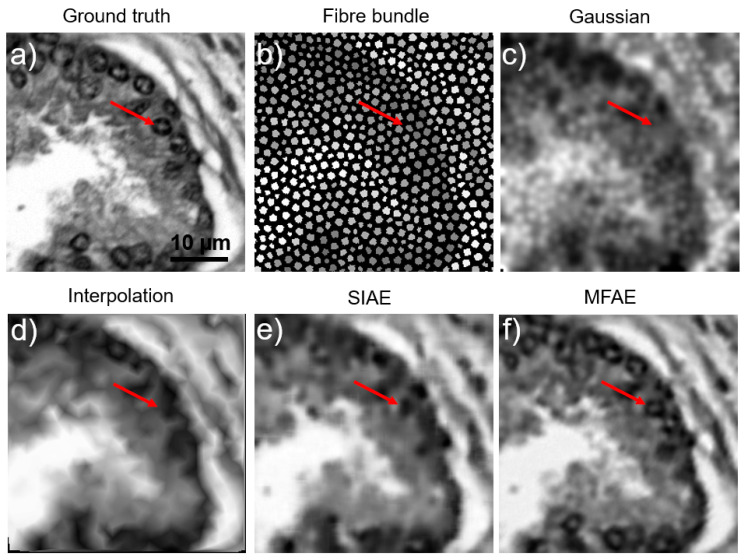
Reconstruction method comparison on acquired prostate images. (**a**) ground truth. (**b**) averaged fiber bundle. (**c**) gaussian blurring. (**d**) linear interpolation. (**e**) single-image autoencoder. (**f**) multi-frame autoencoder. Red arrows point to specific cell nuclei location across each image.

**Figure 6 sensors-23-02469-f006:**
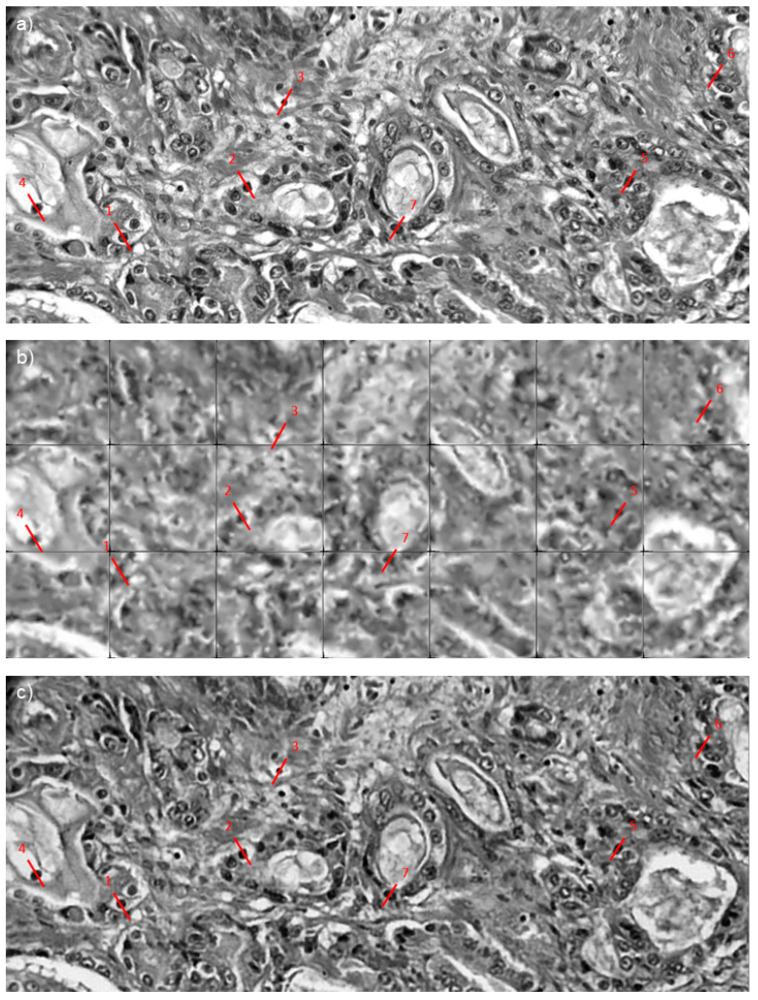
SR reconstruction of slices taken from images prostate tissue (cancer and non-cancer) slide. (**a**) ground truth image. (**b**) Linear interpolation. (**c**) MFAE. Red lines represent line profile taken of cell nuclei, line numbering is equivalent to cell numbers in Table 2.

**Table 1 sensors-23-02469-t001:** Image quality assessment.

	PSNR				SSIM			
	Mean	Max	Min	Std div	Mean	Max	Min	Std div
Gaussian	27.83	28.9	27.3	0.2	0.38	0.66	0.2	0.07
L. Interp	28	31.8	27.3	0.3	0.42	0.81	0.23	0.09
SIAE	27.9	29.1	27.5	0.14	0.32	0.78	0.16	0.08
MFAE	27.73	30.9	26.9	0.59	0.83	0.93	0.66	0.04

**Table 2 sensors-23-02469-t002:** Comparison of cell nuclei sizes *.

Cell Number	Ground Truth	Interpolation	MFAE
1	6.7	5.4	6.7
2	7.0	5.4	7.4
3	4.5	1.3	4.5
4	6.4	5.1	6.4
5	9.3	8.3	9.6
6	7.7	6.1	8.3
7	8	7.4	8.3

* All sizes in µm.

## Data Availability

This was a prospective study, protocol-driven study with ethical approval through the East of Scotland Ethical committee and Caldicott permission (IGTCAL5626) to access the healthcare follow-up data. Also, Tissue request approval was granted from Tayside tissue bank. Machine learning models used in this work were created using standard libraries from TensorFlow and are publicly available on GitHub. https://github.com/MatthewEadie/Square_Prostate_100 (accessed on Tuesday 25 October 2022).

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
