# Peer review of "Fiber Bundle Image Reconstruction Using Convolutional Neural Networks and Bundle Rotation in Endomicroscopy"

_sensors, 2023, doi:10.3390/s23052469_

Round 1

Reviewer 1 Report

In this work, Eadie et al., demonstrated a deep learning-based technology that can achieve super resolution reconstruction of fiber bundle images. The neural network architecture used by the authors is pretty standard, and the overall reconstruction results are quite standard and convincing. However, I still have several technical questions that I would like authors to address.

11. My main concern of this work is that all the fiber bundle images were simulated instead of experimentally collected. While applying a binary mask based on the fiber bundle geometry and spatial averaging are reasonable steps to create synthetic images, I am not sure if that’s enough to generate realistic images. For example, in reality, each fiber can have very different collection efficiency, and is not currently considered in the authors’ model. Also, simply spatial averaging all the pixels within the mask without applying any weighted function (Gaussian window function for example) seems to be insufficient for me. More importantly, I am worried the neural network trained using this method will be heavily dependent on the fiber bundle geometry. If a different mask (different fiber bundle) was used, the reconstruction quality will be affected.

22.  Given that all the H&E-stained slides were from prostate cancer tissue in the study, I suspect that the neural network will only work well on images of similar type of tissue, which is acceptable. But I am curious how sensitivity this network will be to other factors (e.g. FOV, resolution of the system, and fiber bundle geometry as I discussed above).

33.  Surprisingly, the PSNR of the MFAE method is not better than the linear interpolation or SIAE methods, while the SSIM value is significantly better. I would like authors to discuss the potential reason behind that.

44.  The authors characterized the pixel size of the system, but didn’t characterize the lateral resolution of the system. Therefore, I am not sure if the cell nuclei size number reported in table.2 are actually meaningful. For instance, if the resolution of the system is only 1-2um, then any nuclei size difference below that resolution is not necessarily meaningful.

55. The authors explained why 11 images were chosen, which is based on the SSIM metrics. However, from the results presented in Fig.4, the authors didn’t provide the results of 12,13 and 14 channels. Instead the authors only claimed 15 channels resulted in decreased SSIM. To strength authors’ claim that 15 images are the optimal selection, I would like authors to include 12, 13 and 14 channels results.   

66. Another follow-up question. Why did authors choose -10 deg to 10 deg for the rotation range? Is this also an optimal value based on the SSIM metric? Otherwise, my intuition is that a larger angular coverage will be more desired.  

Author Response

Dear Aliya Li and all reviewers,

Thank you for your patience while we made corrections to improve the paper titled “Fibre bundle image reconstruction using convolutional neural networks and bundle rotation in endomicroscopy” manuscript ID: sensors-2035878.

Below we list detailed replies to all points given by each reviewer. As required, the document has the tracked changes turned on to highlight adjusted sections. We have also added comments for each modification to help direct reviewers.

On behalf of all authors, we would like to extend our thanks and appreciation to yourself and each of the reviewers for taking the time to consider this paper and to provide important feedback that will only prove to strengthen this research.

Sincerely,

Matthew Eadie

Reviewer 2 Report

It is a very good topic and can be consider after minor correction 

a- Method need to be more explain, add more detail

b- Result section need to be improve, add more   

Author Response

(The authors gave the same response as above.)

Reviewer 3 Report

This manuscript investigated fiber bundle image reconstruction using deep learning and bundle rotation in endomicroscpy, where convolutional neural networks and multi-frame super-resolution algorithm were developed for the task of interest. The simulated data was used to train and validate the proposed model, with satisfactory results. Overall, the topic of this research is interesting, and the manuscript was well organised and written. The detailed comments are provided as follows.

1.       The contribution and innovation of the manuscript should be clarified clearly in abstract and introduction.

2.       Broaden and update literature review on convolution neural networks or deep learning in engineering applications, such as image processing and data processing. E.g. Vision-based concrete crack detection using a hybrid framework considering noise effect

3.       In general, the performance of CNN model is heavily depenent on the setting of hyperparameters. How did the authors set network hyperparameters in this research to achieve the optimal prediction performance?

4.       Please give more information about the evaulation metrics of proposed method.

5.       The proposed method has not been presented convincingly about its advantages. A comparaive study is suggested to be added via the comparion with other similar methods.

6.       More future research should be included in conclusion part.

Author Response

(The authors gave the same response as above.)

Reviewer 4 Report

Here are my suggestions: In this manuscript, Matthew Eadie et al. proposed a convolutional neural networks for fiber bundle image reconstruction. Below are my questions and concern:
1. Since this work focuses on the improvement to the existing method, I will recommend authors to compare method in the references by J. Shao et al. Optics Letters 44(5) 2019 and J. Shao et al. Optics Express 27(11) 2019. The authors only show SSIM is improved by a factor of 1.97 compared to the method of linear interpolation. Meanwhile, we may not trust SSIM or PSNR, as they calculate globally. This work does not show the advantages of the proposed method compared with the existing method using machine learning.
2. Title of section 2.1 is “Fibre bundle endoscopy system”. The system is actually a microscopic imaging system, not an endoscopic imaging system. Efficiently obtaining accurately matched pairs of raw fiber bundle (FB) images and their “real” GT data is critical for the method of machine learning. A binary mask is placed over the images to obtain raw FB images, which can not represent the real process for FB imaging.
3. The description of training set, verification set and test set in this paper is quite confusing. How to select training set and test set should be described in more detail.

Author Response

(The authors gave the same response as above.)

Round 2

Reviewer 1 Report

Overall, the authors made significant progress in this revision, and I only have 2 minor points that I would like authors to address further. 

1. In my last review, I argued that I am not sure if "cell nuclei size number reported in table.2 are actually meaningful. For instance, if the resolution of the system is only 1-2um, then any nuclei size difference below that resolution is not necessarily meaningful." The authors' response is that they acknowledge the system to only have 3-4 um resolution. Then my question stays the same. If the resolution of the system is at micron level, how would the author be able to report the cell nuclei size down to tens of nanometers?

2. regarding to the question of high sensitivity to fiber bundle geometry, I am OK with authors' response (although I do want to see an unambiguous answer if possible). However, I still want authors to include some discussions about this topic in the main text. 

Reviewer 3 Report

All the technical issues have been addressed.

Author Response

Thank you for your comments.

Reviewer 4 Report

The system is actually a microscopic imaging system, not an endoscopic imaging system. The author should face up to the problems rather than running away from them. For example, title of section 2.1 has been modified, but the problems were not furtherly studied. The reviewer's questions are not considered carefully.

Round 3

Reviewer 4 Report

We suggest that some endoscopic experimental data should be acquired and  processed.
